

# Reanalysis of the anthrax epidemic in Rhodesia, 1978–1984

James M. Wilson[1], Walter Brediger[2], Thomas P. Albright[3] and Julie Smith-Gagen[1]

[1] School of Community Health Sciences, University of Nevada–Reno, Reno, Nevada, United States
[2] Department of Geography, University of Nevada–Reno, Reno, Nevada, United States
[3] Department of Geography and Program in Ecology, Evolution, and Conservation Biology, University of Nevada–Reno, Reno, Nevada, United States

## ABSTRACT

In the mid-1980s, the largest epidemic of anthrax of the last 200 years was documented in a little known series of studies by Davies in *The Central African Journal of Medicine*. This epidemic involved thousands of cattle and 10,738 human cases with 200 fatalities in Rhodesia during the Counterinsurgency. Grossly unusual epidemiological features were noted that, to this day, have not been definitively explained. This study performed a historical reanalysis of the data to reveal an estimated geographic involvement of 245,750 km$^2$, with 171,990 cattle and 17,199 human cases. Here we present the first documented geotemporal visualization of the human anthrax epidemic.

# INTRODUCTION

Anthrax is a potentially lethal disease caused by *Bacillus anthracis*, an aerobic spore-forming bacterium that exists in a complex ecological cycle predominantly involving herbivorous mammals and man. Persistence of anthrax in the environment is due to soil borne spores that remain viable for decades. It is an Office International des Epizooties (OIE) List B disease, for which obligatory reporting is requested of all OIE Member States. The ancient origin of *Bacillus anthracis* is sub-Saharan Africa, specifically the region encompassing Kruger National Park and the North Cape Province of South Africa. Over the centuries, anthrax has been exported throughout the world predominantly through the trade of domestic herbivores (*De Vos & Turnbull, 2004*).

Anthrax was first recognized in colonial times in southern Africa in 1842 and was placed on the list of scheduled diseases in South Africa in 1891. Europeans noted the persistent spread of anthrax in their cattle herds and eventually pushed for control programs that yielded the development of vaccines from 1920 onward. In 1923, South Africa had reported the death of 30,000–60,000 cattle; however, once vaccination was initiated, a dramatic reduction in bovine anthrax was noted. The culmination of an effective vaccination program in South Africa was realized with the Sterne vaccine in 1937, where knowledge of South Africa's success in controlling anthrax was shared with

Corresponding author
James M. Wilson,
jamesmwilson@unr.edu

Rhodesia, now known as the country of Zimbabwe (*De Vos & Turnbull, 2004*; *Mwatwara, 2015*).

In 1978, during the context of the Rhodesian Counterinsurgency, an unprecedented anthrax epidemic in livestock and humans began in Rhodesia. The epidemic progressed largely unchecked until the mid-1980s as the largest known anthrax epidemic in history. This event was documented by Davies in a three-part descriptive study from 1982 to 1985 that was limited to three of five involved provinces (*Davies, 1982*; *Davies, 1983*; *Davies, 1985*). Additional primary source documentation was limited that provided greater insight into the location, case count, spread, and etiology of this epidemic.

A total of 10,738 human cases and approximately 200 deaths (1.9% fatality rate) were reported by JCA Davies in *The Central African Journal of Medicine* across Midlands, Matabeleland, and Mashonaland Provinces from January 1979 through December 1980. The majority of reported human cases were the cutaneous form of the disease; however, all known clinical forms of anthrax infection were documented during the course of the epidemic. The universally acknowledged source of human cases was cattle infected in rural, noncommercial farming areas known as Tribal Trust Lands (*Davies, 1982*; *Davies, 1983*; *Davies, 1985*). The majority, if not all of the cases were among indigenous African farmers living on Tribal Trust Lands. This was an important observation given that approximately 85% of the country was considered agricultural land, of which half was Tribal Trust Lands (*World Bank, 1986*). Because of the extent of agricultural land involved, the epidemic was described as "an economic disaster" (*Turner, 1980*).

The two primary agricultural land classifications of Rhodesia in the late 1970s were Tribal Trust Lands and commercial farming areas. Tribal Trust Lands covered 16.4 million hectares of rural Rhodesia and were designated by the government for subsistence agriculture by indigenous ethnic groups. These areas were known to be of generally marginal agricultural value because of climatic conditions (sporadic rainfall and lengthy droughts), soil quality, and farmers' lack of resources (e.g., lime) to improve and maintain soil quality. Over four million people lived in the Tribal Trust Lands, 800,000 of them belonging to indigenous African farming families; this represented the majority of the rural indigenous population of the country at twice the population density of the Europeans living on the commercial agricultural areas. The majority of cattle raised on the Tribal Trust Lands were used for agricultural labor, personal consumption, trade, and sources of fertilizer. Commercial agricultural areas included farms owned by European farmers who raised cattle for profit, as well as tobacco and other cash crops. The land appropriated for these farms was roughly equal in size to the Tribal Trust Lands and was located in fertile areas (*World Bank, 1986*; *Whitlow, 1980a*; *Whitlow, 1980b*).

African lineages of cattle were comprised of *Mashona, Matabele, Zansi, Amabula, Kavuvu, Amabowe* types and had been present in Rhodesia since at least the early 1800s, which was the limit of documented history in this region of the world. Cattle were an integral part of migrant indigenous peoples of the region during this period. These breeds intermingled with European stock introduced during the colonial period (*Mwatwara, 2015*). Loosely considered, the African lineages resided on the Tribal Trust Lands, whereas the European lineages resided on commercial agricultural lands.

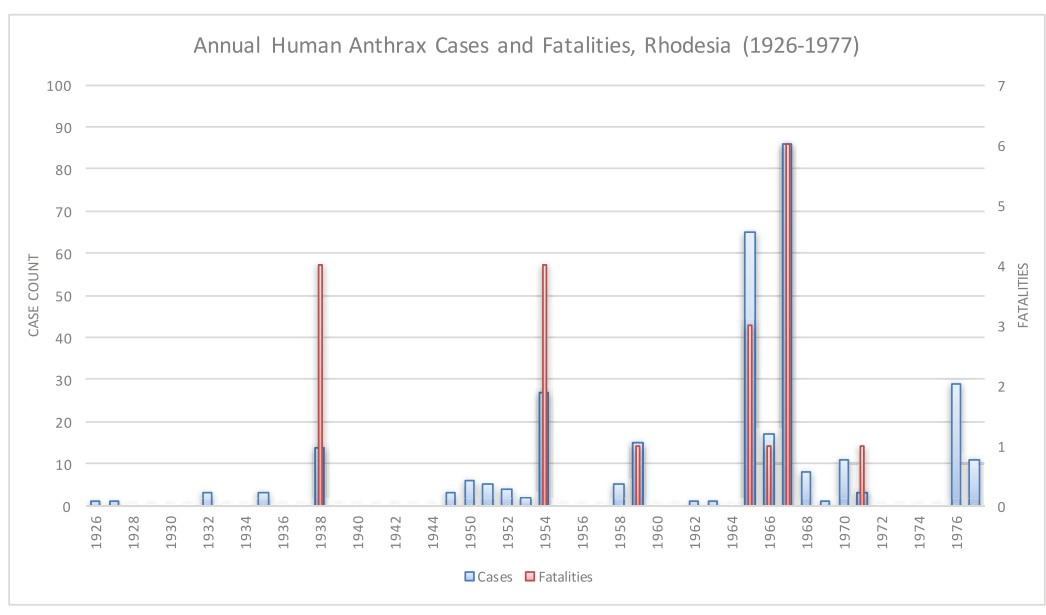

**Figure 1 Historical anthrax human cases reported from 1926 to 1977.** Vaccination for anthrax in cattle was not available until the mid-1950s.

Prior to 1960, Rhodesians had destroyed much of the indigenous wildlife during the expansion of cattle ranching. However, after 1960, concerted effort was directed to the re-establishment of game and wildlife both on private ranches and in national parks. Private ranches engaged in profitable trade with the tanning industry, where elephant, antelope, and zebra skins garnered the bulk of the industry's income in the mid-1970s (*Mossman & Mossman, 1976*).

Prior to the war, the national anthrax control program was considered one of the most advanced and effective in Africa. This was a program that had been in place since approximately the mid-1950s. During the war, vaccination for anthrax was maintained largely on commercial as opposed to Tribal Trust Land farms due to chronic distrust of indigenous Africans directed towards European veterinary practice (*Mwatwara, 2015*). The disruption of standard veterinary services on the Tribal Trust Lands during the war was associated with a dramatic resurgence of multiple diseases in cattle, such as various tick-borne diseases, trypanosomiasis, rabies, and two outbreaks of vaccine-resistant hoof-and-mouth disease (*Davies, 1982*; *Lawrence, Foggin & Norval, 1980*).

Anthrax was rare in Rhodesia prior to 1978, as shown in Fig. 1. In 1898, 41 cattle died of anthrax in Matabeleland Province, which was the first documented appearance of anthrax in Rhodesia. In 1912, 14 pigs and one donkey in Ardbennie and six cattle in Umganin, Bulawayo, died of anthrax. At Mount Hampden, nine cattle died in a limited outbreak in 1917. A larger epidemic in Shamva two years later resulted in the deaths of 102 head of cattle on 18 farms. In 1920, 18 cattle died in Hartley, Mtoko, and Shamva. All of these outbreaks were in western Rhodesia, north of Hartley (*MacAdam, 1980*). Other limited outbreaks in bovines and humans were reported in Chipinga (1952–1954), Mhondoro Tribal Trust Land (1974), and Mount Darwin (1965 and 1970–1971) (*Davies, 1982*; *Davies, 1985*; *Lawrence, Foggin & Norval, 1980*; *Roberts & Chambers, 1975*;
*Mwenye, Siziya & Peterson, 1996*). There were six human cases and two deaths reported in the Mhondoro outbreak; the high fatality rate was attributed to delays in seeking timely medical attention (*Whitlow, 1980b*). Data on human cases were unavailable for the Chipinga and Mount Darwin outbreaks. No outbreaks of anthrax were documented in Rhodesian wildlife until 2004 (*Clegg et al., 2007*).

Human anthrax was rare, as Davies noted, "the majority of doctors in Rhodesia had never seen a case of anthrax" (*Davies, 1985*). The majority of reported human cases in these earlier outbreaks were cutaneous and, to a much lesser degree, gastrointestinal; infections were acquired from handling or eating infected livestock (*Davies, 1982*; *Davies, 1983*; *Davies, 1985*). The approximate annual number of human cases reported nationally was 6 per year, for a total of 322 cases and 20 fatalities from 1926 to 1977 (*Davies, 1982*).

The purpose of this study was to re-examine the 1979–1980 anthrax epidemic in Rhodesia in light of new data and analytic insights gained in the years since this important event.

## MATERIALS AND METHODS

We conducted literature searches using PubMED (United States National Library of Medicine) and AGRICOLA (United States National Agricultural Library) for all references to anthrax in southern Africa from 1970 to the present. We also reviewed all available veterinary and agricultural literature published in Rhodesia from 1975 to 1985. Historical land classification maps published for Rhodesia from 1975 to 1981 were obtained from the United States Library of Congress (*Surveyor-General, 1978*). All available original manuscripts regarding anthrax in Rhodesia were reviewed, and a reanalysis of the epidemiological data contained therein was performed. We examined references to epidemic features such as historical (pre-epidemic) data, route of infection, severity of disease, meteorological data and seasonality, host animals, potential vectors, and vaccination coverage. The sources of these different aspects of the epidemic are referenced in the results section.

World Meteorological Organization archival average temperature and precipitation data for Rhodesia was used to assess meteorological anomalies for 1977–1980 (*World Bank, 2015*). Anomalies were calculated using the following equation:

$$anomaly \ = \ \frac{(monthly \ value \ - \ monthly \ average)}{monthly \ standard \ deviation};$$

Where monthly data for 1901–1976 was used to calculate the monthly average and standard deviation.

In order to assess the spread of anthrax outbreaks in Rhodesia in space and time, we performed a simple geospatial analysis using ArcGIS (ESRI, Redlands, CA, USA). We defined an "outbreak" as an incipient focus of human anthrax cases that appeared in a district level hospital, which was the finest spatial resolution available in the data. The month and year of first cases reported to these district hospitals was the only data available; continuous monthly times series case counts were not available. We georeferenced hospital locations using Wikipedia (https://www.wikipedia.org) and

Google Maps (https://www.google.com/maps). Because nearly all of the human anthrax cases followed contact with cattle exhibiting signs of anthrax infection, we investigated the availability of bovine anthrax data from 1978 to 1980. We were unable to identify any surviving record of bovine anthrax data for this time period (S.M. Chikerema, 2016, personal communication).

To evaluate JCA Davies' description of contiguous and non-contiguous geotemporal spread, we then calculated the centroid (arithmetic mean) of outbreak locations for each time step in our series (*Davies, 1982*; *Davies, 1983*; *Davies, 1985*). Geospatial cluster analysis was performed using the Kulldorff space-time permutation model resident in SaTScan v9.4 (*Kulldorff, 2015*; *Kulldorff et al., 1998*), where the objective was to statistically identify space-time clusters such as reports of multiple waves of human cases within the context of the epidemic itself. Default settings were used for the analysis to allow for identification of possible seasonal patterns.

## RESULTS

First recognition of the epidemic was in Nkai District, Matabeleland Province, November 1978, with a low number of human cases reported until June 1979. All of the cases were associated with the butchering and skinning of local cattle. Nkai Hospital would later report over 500 cases from January 1979 to October 1980, which was 1.5 times that of all human cases reported from 1926 to 1977. Approximately half of these cases required hospitalization, with 17 fatalities (case fatality rate 3.3%). Eight of the fatalities were due to respiratory anthrax (47%). The remainder died of sepsis that followed a cutaneous lesion. This was considered the first phase of the epidemic (*Davies, 1982*; *Davies, 1983*; *Davies, 1985*).

The epidemic smoldered until mid-1979, when a second phase of the epidemic was apparent. Case counts abruptly increased from September to November 1979 (Fig. 2) ahead of the rainy season peak in December–January (Fig. 3) (*World Bank, 2015*). While extreme meteorological phenomena were not reported previous to or during the time period of the epidemic, higher than normal precipitation occurred in early 1978 (Fig. 4) (*World Bank, 2015*). Geotemporal variation in epidemic progression was observed at the provincial level (Fig. 5), where epidemic peaks were sequential across Matabeleland (November 1979), Midlands (December 1979), and Mashonaland Provinces (February 1980). Midlands Province was noted to begin peaking in October 1979 (Fig. 6) (*Davies, 1982*; *Davies, 1983*; *Davies, 1985*). This was suggestive of a general southwest to northeast progression. The patients were dominantly Tribal Trust Land inhabitants, with rare documentation of cases among those in the tanning industry (*Davies, 1983*; *Davies, 1985*). Over the course of five months (November 1979–March 1980), explosive spread of anthrax in Mashonaland Province had encompassed 13 districts and 120,000 km$^2$ (*Davies, 1982*; *Davies, 1983*; *Davies, 1985*).

The Beatrice Road Infectious Diseases Hospital, located in Salisbury (now called Harare), saw 712 cases from January 1980 to June 1982 (Fig. 7). These patients were referred to the hospital from across Mashonaland Province. All were Tribal Trust Land inhabitants except one individual from a tanning facility. Resource strain at the hospital
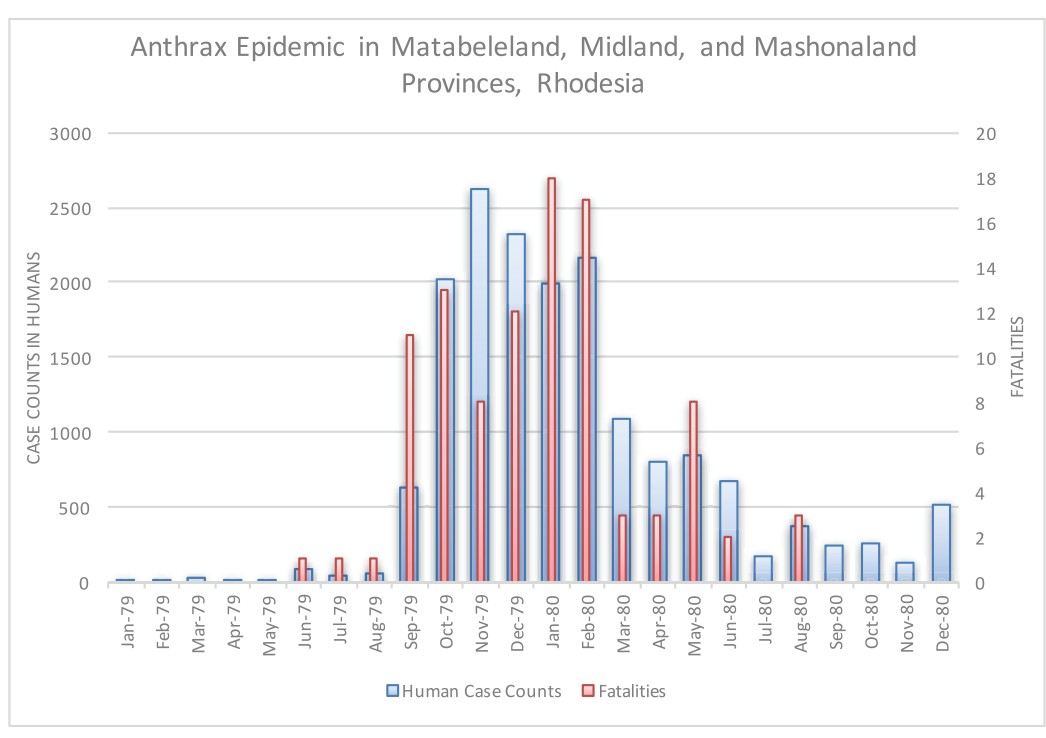

**Figure 2 Rhodesian human anthrax cases from January 1979 to December 1980 for the provinces of Matabeleland, Midland, and Mashonaland.**

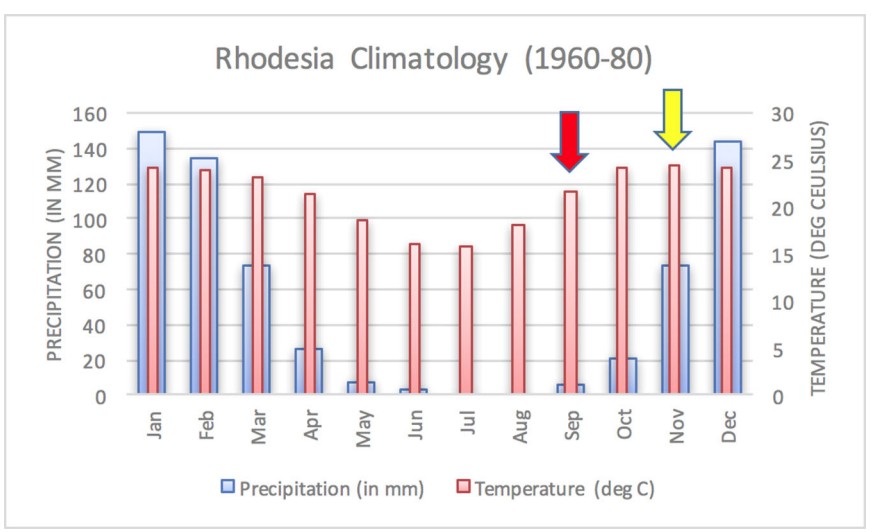

**Figure 3 Average annual Rhodesia climatology based on data from 1960–1980.** A dramatic surge in human cases was observed ahead of the peak of the rainy season. Arrows indicate the start of the first (yellow) and second (red) phases of the human epidemic, in November 1978 and September 1979, respectively.

was reflected in demand for hospitalizations, as noted in February 1980 and December 1980, which was the result of two waves of patients who were hospitalized for two to five weeks. By March 1980, the Beatrice Road Infectious Diseases Hospital reported that the

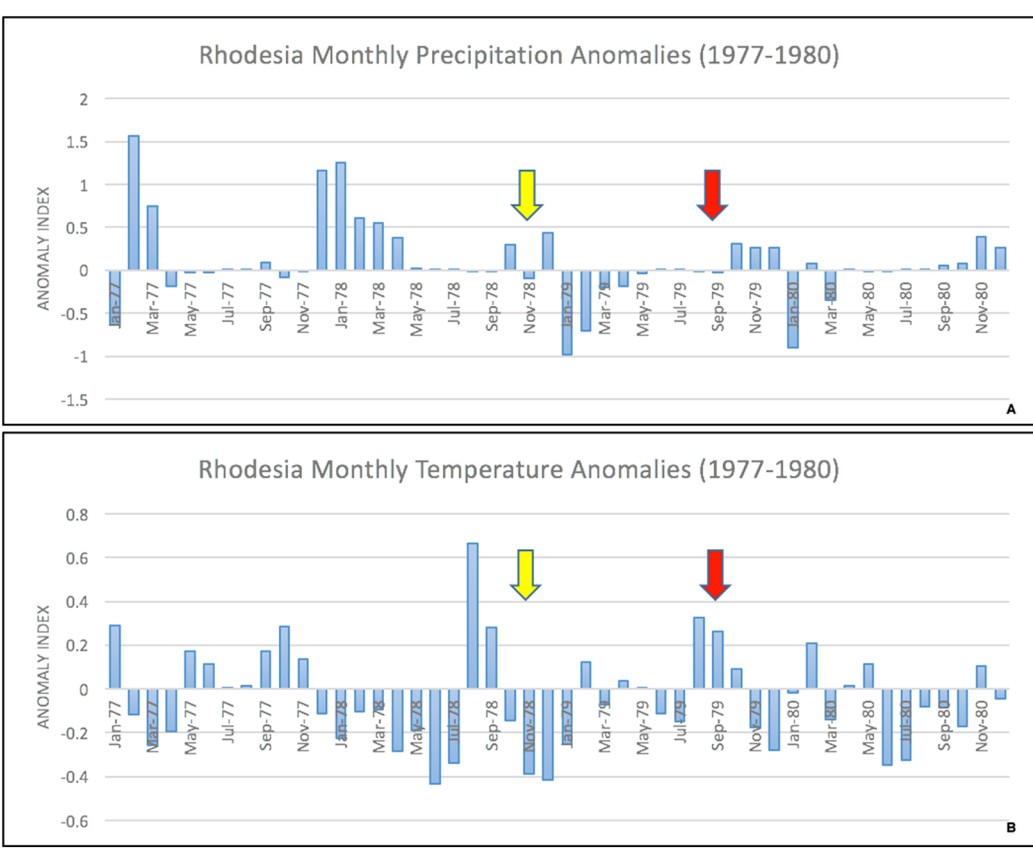

**Figure 4 Monthly temperature and precipitation anomalies for Rhodesia, 1977–1980.** Anomaly calculations used data from 1901–1976 (see Materials and Methods). Arrows indicate the start of the first (yellow) and second (red) phases of the human epidemic, respectively.

most common reason for admission to their hospital was anthrax. The number of cases seen at Beatrice Road was considered enough to prompt concerns about contamination of the hospital with anthrax spores, which resulted in heavy use of masks, gowns and gloves until supplies were unable to meet the demand. However, no infection was noted among healthcare providers or between patients. Although many medical facilities reported abrupt, significant strain on their resources, fatality rates were considered very low and manageable. There was no report of antimicrobial resistance; rather, the vast majority of the patients were effectively managed with penicillin. There was no indication of a penicillin shortage (*Davies, 1985*).

Inhalation, gastrointestinal, and meningitis presentations were documented at the Beatrice Road Infectious Diseases Hospital as well as additional hospitals (*Davies, 1982*). Five cases of anthrax meningitis, all fatal, out of 18 total cases seen were reported over the course of 12 months at Parirenyatwa General Hospital, also in Salisbury. This facility had not previously seen a single case of anthrax from 1970 to mid-1979. All of these fatal cases had cutaneous lesions that followed contact with cattle upon presentation. Several of these cases had reported a painful insect bite that preceded development of the classic anthrax eschar lesion. Death was observed within one week of initial symptoms. This experience prompted several case report publications because these presentations

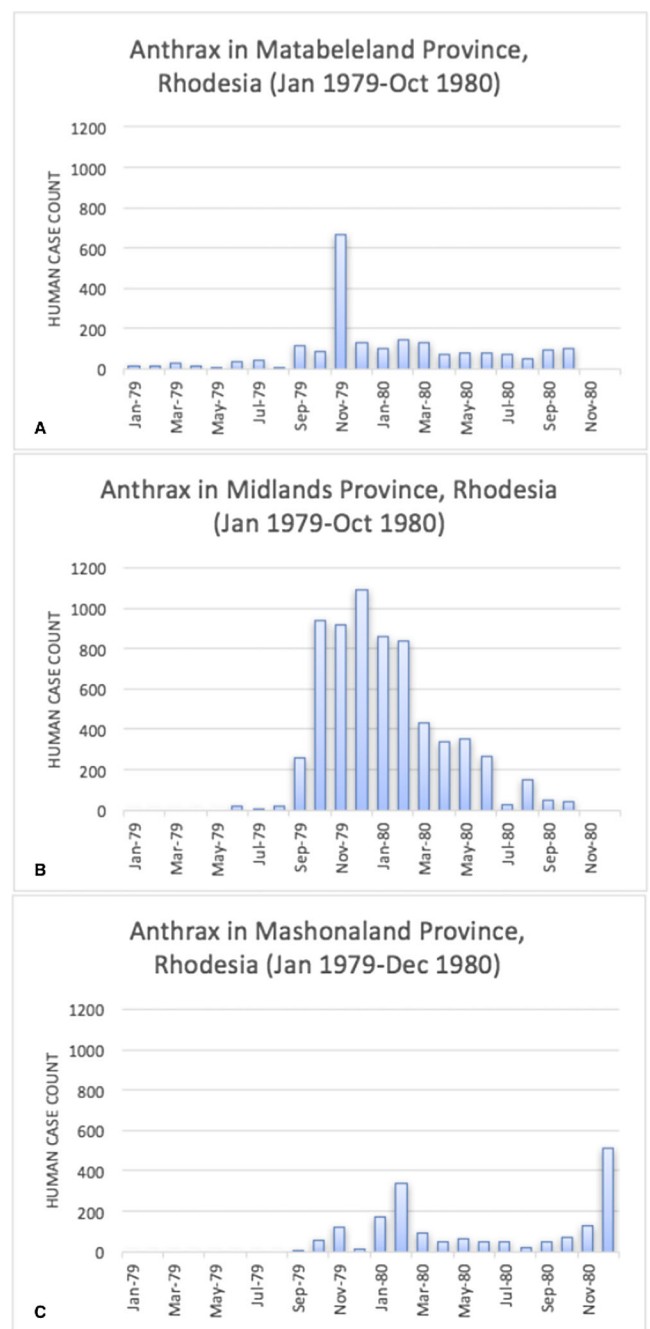

**Figure 5 Human anthrax cases by province, noting the temporal shift in the peak of cases as the epidemic progressed across the country.** A second wave of cases is observed in the data for Mashonaland Province.

had previously been "extremely rare" in Rhodesia (*Davies, 1985*; *Levy et al., 1981*; *Latif & Nathoo, 1983*; *Jena, 1980*). The unusual volume and variety of clinical presentations led to comparisons at the time to the Sverdlovsk, U.S.S.R. outbreak of anthrax (*Turner, 1980*). The Sverdlovsk outbreak was later shown to be an accident of a biological weapons laboratory (*Meselson et al., 1994*).

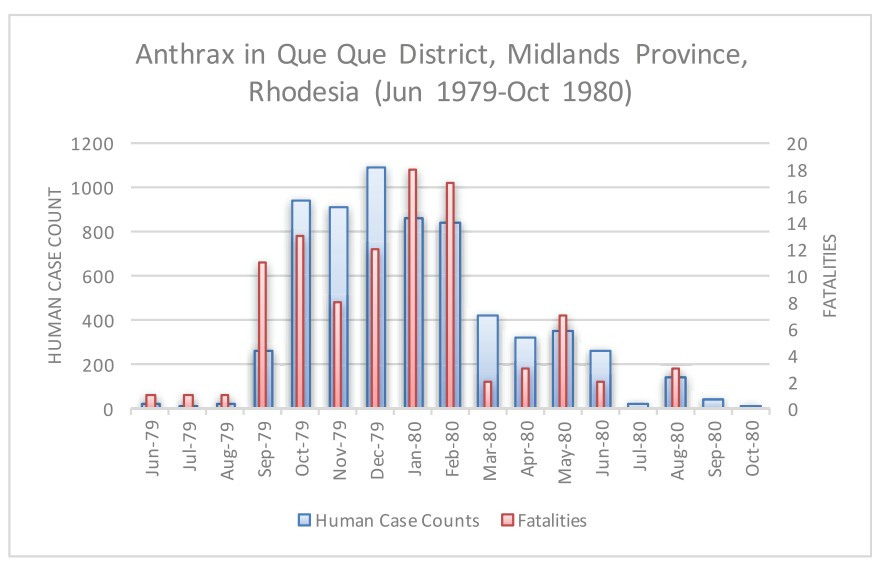

**Figure 6 Human anthrax cases in Que Que District, which experienced a prolonged peak from October 1979 to February 1980.**

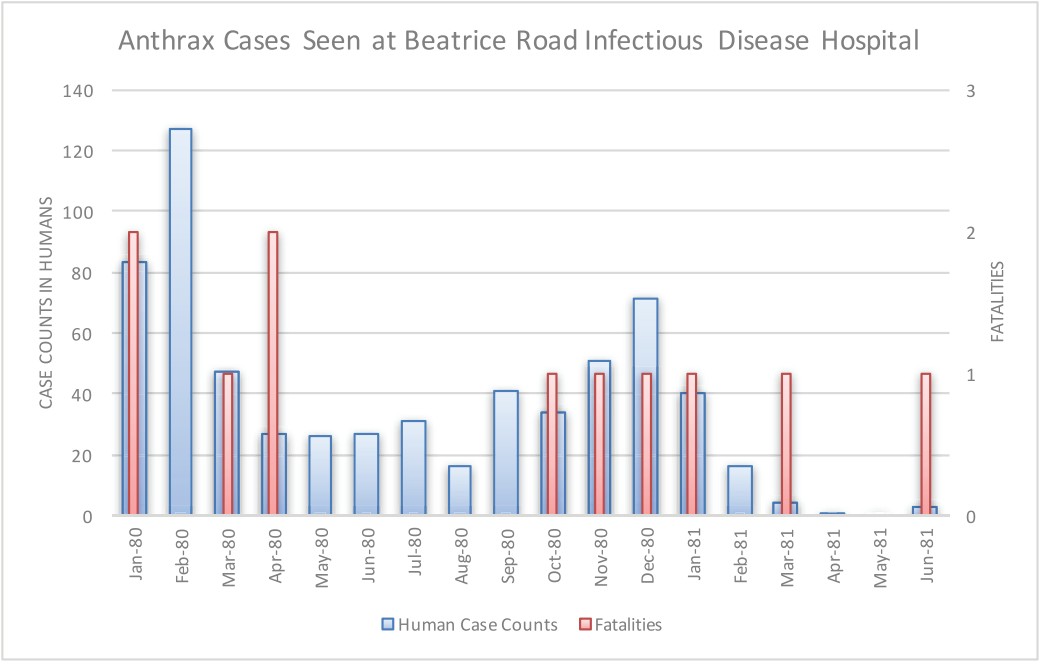

**Figure 7 Human anthrax cases seen at the Beatrice Road Infectious Disease Hospital.** This hospital, by March 1980, reported that anthrax was the leading reason for hospital admission. The facility experienced resource strain in two periods that coincided with two waves of patients in February 1980 and again in December 1980.

Figure 8 displays where human cases were first recognized, at the finest temporal and spatial resolution the epidemiological data allowed (i.e. monthly and district hospital coordinate). Time series data for case counts for district level hospitals was unavailable; therefore data regarding multiple epidemic waves seen for a given hospital

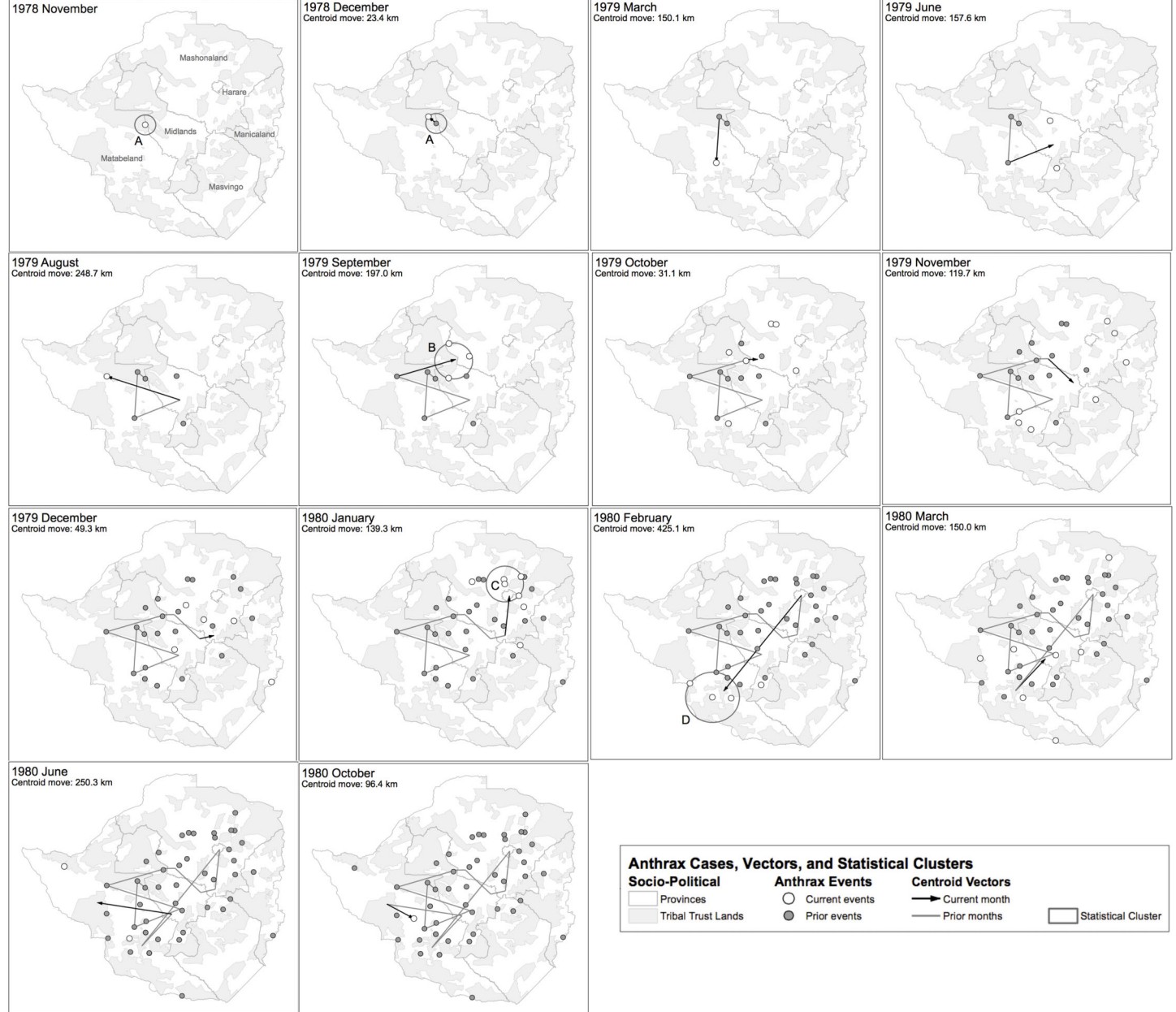

**Figure 8 Geospatial time series progression of the anthrax epidemic in humans.** The mean of outbreak coordinates is indicated as a centroid coordinate, where each time step is associated with spatial movement of that coordinate. Only time steps associated with the first human cases for each site are shown. (A–D) Geotemporal clusters associated by the Kuldorff statistic with higher relative risk for high anthrax case counts are shown and denoted with a capital letter. According to Davies, the spread of anthrax in humans involved a non-contiguous, heterogeneous distribution pattern.

was unavailable with the exception of the Beatrice Road Infectious Disease Hospital (Fig. 7). After June 1979, the epidemic spread eastward to Que Que District, Midlands Province, then erratically in the context of a "puzzling hop[ping]" phenomenon, whereby anthrax cases appeared in multiple foci with non-contiguous involvement of the land between (Davies, 1982). We attempted to further elucidate the hopping phenomenon

**Table 1 Results of the Kuldorff space-time permutation analysis.** Statistical significant was defined by P < 0.05. The clusters identified were associated with non-significant, higher relative risk (*Kulldorff, 2015*).

| Cluster | Coordinates | Radius | Time frame | No. of outbreaks | Expected outbreaks | Observed/ Expected | Test statistic | P-value |
|---|---|---|---|---|---|---|---|---|
| A | 18.999396 S, 28.901210 E | 32.95 km | Nov-78 to Dec-78 | 2 | 0.087 | 23 | 4.4 | 0.24 |
| B | 18.469056 S, 29.444797 E | 59.37 km | Sep-79 | 3 | 0.26 | 11.5 | 4.67 | 0.055 |
| C | 17.507240 S, 30.975851 E | 58.94 km | Jan-80 | 4 | 0.61 | 6.57 | 4.27 | 0.25 |
| D | 20.916667 S, 28.466667 E | 83.34 km | Feb-80 | 3 | 0.43 | 6.9 | 3.3 | 0.79 |

by using a combination of centroid distance measurements and the Kuldorff statistic. The Kuldorff space-time permutation highlighted four clusters of non-significant, but increased relative risk for high anthrax activity as shown in Table 1 and displayed in Fig. 8. The map corresponding to November 1979 in Fig. 8 is an example of a hop, which involved non-contiguous spread to Filabusia, Umzingqane, and Bembezi Districts from Nkai and Que Que Districts. Another example of a hop was in March 1980, with spread to Chilimanze, Charter, and Seluweke. These hops involved distances of 40–50 km. According to Davies, "The intervening commercial and communal areas did not report any cases at this time." The context of these observations was during the second phase of the epidemic, when explosive spread was noted to involve 120,000 km$^2$ within five months in Mashonaland Province (*Davies, 1982*).

Up to the point of the epidemic, an average of 20 cases were seen in livestock annually (*Lawrence, Foggin & Norval, 1980*). In the pre-1980 time period, systematic animal disease surveillance was not performed, especially on the Tribal Trust Lands. The epidemic among humans began nearly exclusively through contact with cattle across all areas of involvement in Rhodesia. In Lupane, within the epicenter, more than 5,000 head of the total cattle population (5%) had died (*Lawrence, Foggin & Norval, 1980*; *Kobuch et al., 1990*). National statistics for the total number of anthrax-related cattle deaths from 1979–1980 were unavailable, but one report cited "many thousands" (*Lawrence, Foggin & Norval, 1980*). Up to 50% of cattle herds died in some communities (*Anonymous, 1979*). After conducting a full literature review, we were unable to document reports of anthrax in any of the neighboring countries during the time of the Rhodesia epidemic. There was no report of any other animal species involved in the epidemic, with the exception of the rare, occasional domestic goat (*Kobuch et al., 1990*). The first documented appearance of anthrax in the wildlife of Zimbabwe was in 2004 (*Clegg et al., 2007*). Because no point source could be identified, introduction of anthrax to cattle via feed or fertilizer was hypothesized; however, rejected in favor of emergence from endemic foci (*Kobuch et al., 1990*). Access to safe drinking water on the Tribal Trust Lands for both humans and cattle became a serious concern (*Davies, 1983*).
Chikerema et al. (2012) recently examined archival bovine anthrax data for Zimbabwe dating back to 1967; however, it was a acknowledged there was no data on bovine anthrax for the period examined in this study (S.M. Chikerema, 2016, personal communication). From 1967 to 1976, 12 districts had reported bovine anthrax, followed by dramatic expansion to 42 districts by 2006. The majority of bovine anthrax continues to be documented in rural areas. As of 2006, one of the districts associated with the highest risk of bovine anthrax activity was also identified by this study as higher relative risk, but not statistically significant, as Cluster A (P = 0.24). This likely reflects a difference in anthrax zones in the late 1970s versus expansion of anthrax zones observed by 2006.

Chikerema et al. (2012) also identified seasonality at the national level in bovine anthrax activity where a peak was noted in October that coincides with peaking seasonal temperature and the beginning of the rainy season (Fig. 3). Figure 2 shows the approximate national view of the human case counts summarized over three of the five provinces that reported cases. The cases peaked in November, the majority of which followed contact with infected cattle. The epidemic began in humans in November 1978. The Beatrice Road Infectious Disease Hospital had observed two waves of patients, in February and December, respectively. The Kuldorff statistical analysis identified November 1978 (Cluster A, P = 0.24), September 1979 (Cluster B, P = 0.055), and January and February 1980 (Clusters C and D, P = 0.25 and 0.79, respectively). Our findings suggest higher relative risk for human anthrax cases followed the period identified by Chikerema et al. (2012) as seasonal peaking bovine anthrax activity.

During the course of the epidemic, veterinary control measures had waned in the context of the Rhodesian Counterinsurgency. Anthrax vaccines were acquired in response to the epidemic, but supply was limited (Lawrence, Foggin & Norval, 1980). The response campaign was considered ineffective due to poor utilization by the effected communities, due to pressure from insurgent forces to not cooperate with the national veterinary service (Anonymous, 1979). After the war in 1981, vaccination services resumed and reached 70% of the national cattle herd by 1985 (Lawrence, Foggin & Norval, 1980). Despite re-establishment of control, bovine anthrax activity continued to expand dramatically to 2006 (Chikerema et al., 2012).

Substantial underreporting of human cases by as much as 50% was inferred given the number of district hospitals and medical facilities who responded to Davies' survey and the likelihood that many patients either chose not to be evaluated by a medical facility or did not survive long enough for evaluation, as was observed in prior anthrax outbreaks in Rhodesia (Davies, 1982; Roberts & Chambers, 1975). Ongoing conflict over much of the anthrax-involved areas also impeded communications (Cilliers, 1985). This represents an important source of potential bias when examining the results of the space-time permutation model described above.

We found that Davies' text description of the total human cases equating to 10,738 to be in discrepancy with the epidemiological data presented in his manuscripts' tables for Midlands, Matabeleland, and Mashonaland Provinces from January 1979–December 1980, which collectively reported a total of 17,068 cases and 101 fatalities (Davies, 1982).

In terms of overall case counts for the epidemic, we found this to underestimate the true extent of the epidemic as well, given:

- Data reported by Davies did not include summary statistics for the involved areas of Manicaland and Victoria Provinces; however, it was briefly discussed in his Beatrice Road manuscript (*Davies, 1982*; *Davies, 1983*; *Davies, 1985*);
- Beatrice Road Infectious Diseases Hospital in Harare, Mashonaland Province reported 131 additional cases and four fatalities (0.03% fatality rate) from January 1981–June 1982 (*Davies, 1985*);
- *Kobuch et al. (1990)* reported anthrax cases in Lupane extending through 1984; and
- Statistics reported by Davies for Mashonaland Province indicated a second epidemic peak in December 1980 without evidence of transmission resolution (i.e. incomplete data) (*Davies, 1982*; *Davies, 1983*; *Davies, 1985*).

Based on inclusion of this new data we found an overall discrepancy of 10,738 versus 17,199 cases and over 200 fatalities versus 105 fatalities if only the data tables from Davies' manuscripts are considered. We were unable to account for those patients on the rural Tribal Trust Lands who were infected, unable to seek appropriate medical care, and be documented in the epidemiological reports. A larger number of subclinical cases and exposures were likely based on observations of the low relative infectivity of anthrax spores in humans (*Turnbull, 1998*). Compared to annual baselines prior to this epidemic, this was at least more than 1,400 times above expected annual baselines in the half-century prior to this epidemic. Based on the observation that the majority of these human cases were Tribal Trust Land residents, we estimate the attack ratio to have been 17,199 cases in the rural farming population of 800,000 (2.1%) versus the overall Tribal Trust Land population of at least four million (0.43%). Based on an estimated one human cutaneous case to 10 infected cattle carcasses, we estimate the true number of infected cattle to be 171,990 (*Turnbull, 1998*). When attempting to discern the duration of the event, it is apparent the epidemic began in November 1978 and extended beyond 1980, to at least December 1984. This too raises questions regarding additional unreported cases that occurred beyond the purview of Davies' original reports. Overall, we estimate the epidemic covered approximately 245,750 km$^2$, using a minimum convex polygon around all outbreak locations in our dataset. We acknowledge the actual land area was likely less due to the non-contiguous spread reported by Davies, where true involvement of the land was not spatially contiguous.

## DISCUSSION

The unusual, unprecedented nature of the epidemic and how it became manifest in Rhodesia was the subject of several papers. Davies emphasized the pattern of case counts in relation to seasonal precipitation and tabanid (horsefly) counts (Fig. 9) (*Phelps & Vale, 1975*) as a way of explaining the abruptness and magnitude of the epidemic's spread possibly due to a combination of flooding and biting arthropods (*Davies, 1985*). This hypothesis was supported by multiple accounts by patients who received a painful

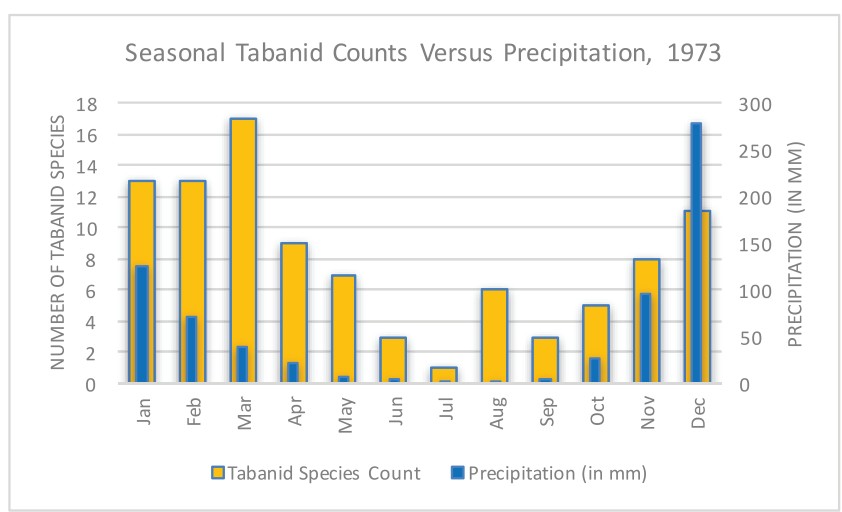

**Figure 9** Seasonal tabanid species counts versus precipitation in 1973 (*World Bank, 2015*; *Phelps & Vale, 1975*). This data reflected a limited study of seasonal tabanid density and became a focus of investigation to explain the rapid and pervasive spread of anthrax in Rhodesia.

insect bite that later evolved into an eschar (*Davies, 1983*). There were no available observations of tabanids biting cattle; however, this may be assumed because tabanid blooms coincided with the appearance of bovine anthrax (*McKendrick, 1980*). There was no report of extreme meteorological conditions or tabanid activity to explain the vast geography of transmission. This hypothesis has been met with a high degree of debate over the years, with *Kobuch et al. (1990)* remarking in 1989:

> In Zimbabwe, the Tabanidae multiply seasonally during the rainy summer months (October to April). They will settle and feed on carcasses of dead animals or on open wounds of the living and they readily bite humans, horses, cattle, and other livestock. The data for the tabanids derive from a detailed study carried out in 1973 and it has to be assumed that similar patterns occur annually. While the rise-fall-rise patterns of tabanid species counts parallel the numbers of anthrax cases, the concept of an association between biting flies and the incidence of anthrax remains a subject of controversy.

In 2004, blowflies were later shown to be an important vector in Rhodesia during the context of a multi-species epidemic. The evidence that supported this hypothesis was the observation that browsers (kudu) versus grazers were predominantly infected. This was due to blowflies feeding on anthrax-infected carcasses and then regurgitating live *Bacillus anthracis* on leaves at grazing height versus directly on the ground (*Clegg et al., 2007*). Davies suggested the possible involvement of vultures in the transmission cycle, but this hypothesis was not supported by contemporary studies (*Davies, 1985*; *Mundy & Brand, 1978*).

An alternate hypothesis was proposed by Meryl Nass, an American physician living in Zimbabwe at the time who suggested the epidemic was propagated intentionally. She emphasized the unusual features of the epidemic: large numbers of cases, geographic

extent and involvement of areas that had never reported anthrax before, lack of involvement of neighboring countries, specific involvement of the Tribal Trust Lands versus European-owned agricultural land, and coincidence with an ongoing civil war (*Nass, 1992*). Witness testimony from Tribal Trust Land inhabitants revealed a belief that "poisoning" by anthrax occurred during the Counterinsurgency (*Alexander, McGregor & Ranger, 2000*; *White, 2004*). Other authors provided testimony of deliberate anthrax releases during the Counterinsurgency by Rhodesian and South African forces and speculated this activity was a progenitor to South Africa's Project Coast, a biological weapons program (*Gould & Folb, 2002*; *Burgess & Purkitt, 2001*). Despite others supporting Nass' provocative debate (*Mangold & Goldberg, 1999*; *Martinez, 2002*), no additional evidence has been offered to support this assertion.

A review of Fig. 8 shows the epidemic generally avoided what Rhodesian forces considered Vital Asset Ground during the Counterinsurgency; however, this observation was not universal. Vital Asset Ground was generally defined as the center ovoid area of the country consisting of non-Tribal Trust, European-owned land that ran from the southwest to the northeast. Joseph Nkomo's Soviet and Cuban-backed ZIPRA armed forces were based to the west of Rhodesia, and began an escalation of attacks into Matabeleland Province in 1978. In September 1978, a Soviet ground missile launched by ZIPRA forces downed a civilian Air Rhodesia Viscount. Rhodesian media had reported that 10 of 18 survivors were massacred on the ground by ZIPRA forces after the plane crashed, provoking a dramatic escalation in the conflict. Soon thereafter, Maoist ZANU forces led by Robert Mugabe in turn escalated their attacks to the east in Mashonaland Province. All of Rhodesia was subsequently involved in conflict with ZANLA and ZIPRA guerilla forces that expanded their infiltration of the country from 8,952 to 11,183 personnel (25% increase) from December 1978 to January 1979. Infiltration routes spanned nearly every sector of border in the country during this period and exploited access to the resources and support of the Tribal Trust Lands as the insurgents ranged inwards towards the center of the country (*Cilliers, 1985*). The first phase of the epidemic of anthrax began in November 1978 in Nkai, Matabeleland Province. The second phase of the epidemic, which was focused on Mashonaland Province, escalated dramatically in late 1979 (*Davies, 1982*; *Davies, 1983*; *Davies, 1985*). The Kuldorff statistic identified clusters of high anthrax activity in November and December 1978, September 1979, and January and February 1980.

Bovine and human anthrax had been reported in years past in Nkai, Que Que, and many other districts involved in the epidemic (*Davies, 1982*). There are notable observations, however, that confound an explanation regarding the unusual nature of this epidemic within the context of modern history:

- There was no report of anthrax in wildlife in Rhodesia/Zimbabwe until 2004 (*Clegg et al., 2007*).
- No point source of anthrax was identified (*Kobuch et al., 1990*).
- As of the date of our study, the volume of human anthrax cases reported here has not been previously reported anywhere in the world.

- The scale of cases in cattle and humans and associated morbidity and mortality was suggestive of low indigenous herd immunity (*De Vos & Turnbull, 2004*; *Cizauskas et al., 2014*).

- Anthrax was previously not observed to transmit so pervasively in Rhodesia as far back as 1898, when neither antibiotics or vaccines were available (*MacAdam, 1980*). This is despite the well documented presence of both cattle and humans dating back to the early 1800s (*Mwatwara, 2015*).

- There has, to-date, been no recorded instance in global history of a combined bovine-human anthrax epidemic of this magnitude attributed to an interruption of a cattle vaccination control program.

- The observation of all clinical forms of anthrax within such a small time period has not been previously documented in world history (*Davies, 1982*; *Davies, 1983*; *Davies, 1985*; *Turner, 1980*).

- There were no reports of unusual population densities of tabanid or other families of candidate vector arthropods present at the time (*Davies, 1985*).

- There were no reports of grossly unusual meteorological activity at the time (*World Bank, 2015*; *Kobuch et al., 1990*; *Unganai & Mason, 2001*).

- There was no confirmatory evidence to explain the apparent geotemporal and non-contiguous spread observed with cattle and human cases reported during the epidemic.

- Multiple physicians, veterinarians, public health and biodefense professionals focused their attention on an explanation of the unusual epidemiological features of this event (*Davies, 1982*; *Davies, 1983*; *Davies, 1985*; *Turner, 1980*; *Lawrence, Foggin & Norval, 1980*; *MacAdam, 1980*; *Levy et al., 1981*; *Latif & Nathoo, 1983*; *Jena, 1980*; *Kobuch et al., 1990*; *McKendrick, 1980*; *Nass, 1992*).

- Multiple independent sources reported the use of chemical and biological weapons by Rhodesian forces (*Nass, 1992*; *Alexander, McGregor & Ranger, 2000*; *White, 2004*; *Gould & Folb, 2002*; *Burgess & Purkitt, 2001*; *Mangold & Goldberg, 1999*; *Martinez, 2002*). The appearance and clustering of anthrax in humans geotemporally mirrored escalation in conflict with ZIPRA and ZANLA forces (*Cilliers, 1985*).

The sole possible exception to some of the above observations is the purported anthrax epidemic of 1770 in Haiti (then-Saint Domingue) that killed 15,000 indigenous and European people over a period of six weeks. The dominant clinical form of anthrax in Haiti was hypothesized to be gastrointestinal following the distribution of contaminated meat (*Morens, 2002*). In comparison, the Rhodesia epidemic occurred over a longer time frame, with involvement of non-contiguous areas, and with several waves of transmission (see Fig. 8).

Contiguous, localized spread of anthrax in humans was observed to be due to a stressed African indigenous population slaughtering their cattle for food and engaging in local commerce with the contaminated meat (*Davies, 1982*; *Davies, 1983*; *Clegg et al., 2007*). This was a contributing factor in the context of food insecurity during the war.

The areas of anthrax involvement spanned multiple indigenous tribes across areas of intense conflict where the roadways were mined and travel was restricted for the general public. ZANLA and ZIPRA forces often fought each other as well as Rhodesian forces and were supported by different indigenous communities (*Clegg et al., 2007*). In all of the district hospitals, reported human cases had slaughtered local cattle prior to their illnesses, where the majority of cases were initially cutaneous presentations (*Davies, 1982*; *Davies, 1983*; *Davies, 1985*). An apparent high fatality rate among cattle is suggestive of either exposure through inhalation or gastrointestinal routes (*De Vos & Turnbull, 2004*). It remains unclear how the cattle were infected across such vast geography. It may be hypothesized that transport of live cattle, as was often observed when indigenous African owners sought to avoid official scrutiny during peacetime, could have also contributed to spread. We propose this was unlikely to account for the distances of the different novel foci that appeared in the various locations of the country due to a combination of (1) tribal boundaries and ownership and (2) restrictions to population movement due to land mines and enforcement of movement controls by authorities during the war. Therefore, while local trade of infected meat or cattle movement likely played some role in human exposures, it does not explain the spread of anthrax across such vast geography.

In summary, one hypothesis is emergence and subsequent expansion of anthrax from a combination of endemic ecological foci and increasing cattle population density in the context of poor veterinary control practice from 1967 to 2006 (*Chikerema et al., 2012*). Expansion was facilitated by a combination of seasonal meteorological factors such as temperature and precipitation, along with possible inclusion of arthropod vectors. Under this hypothesis, anthrax' expansion was an unfortunate outcome of the collapse of control measures during the Counterinsurgency from which Zimbabwe has yet to fully recover.

The competing hypothesis is anthrax was endemic in Rhodesia prior to the Counterinsurgency; however, deliberate exploitation of the wartime environment through intentional release of anthrax occurred at various times and locations in Rhodesia. Under this hypothesis, new endemic foci were created after cattle were targeted by Rhodesian or South African forces. The collapse of control measures enabled further spread, which was enhanced by seasonal meteorological factors with or without arthropod vectoring. Current activity of anthrax in Zimbabwe now follows endemic seasonal patterns.

No definitive explanation to-date has been offered to explain the many unusual features of this epidemic. We acknowledge that lack of data regarding several of the observations related to this epidemic provides important sources of bias in this study. Despite the limitations of the data associated with this important historical event, and in agreement with other authors (*Chikerema et al., 2012*), we propose a need for further scrutiny of the etiology of this epidemic, which contributed to substantial human suffering and the destruction of a once-vibrant national agricultural economy.

## ACKNOWLEDGEMENTS

The authors would like to thank JCA Davies and GA Cross for their input and the reviewers of this manuscript for their consideration.

### Funding

The University of Nevada-Reno School of Community Health Science financially supported this study. The funders had no role in study design, data collection and analysis, decision to publish, or preparation of the manuscript.

### Grant Disclosures

The following grant information was disclosed by the authors:
University of Nevada-Reno School of Community Health Science.

### Competing Interests

The authors declare that they have no competing interests.

### Author Contributions

- James M. Wilson conceived and designed the experiments, analyzed the data, contributed reagents/materials/analysis tools, wrote the paper, prepared figures and/or tables, reviewed drafts of the paper.
- Walter Brediger performed the experiments, contributed reagents/materials/analysis tools, prepared figures and/or tables, reviewed drafts of the paper.
- Thomas P. Albright performed the experiments, contributed reagents/materials/analysis tools, wrote the paper, reviewed drafts of the paper.
- Julie Smith-Gagen analyzed the data, reviewed drafts of the paper.

### Data Deposition

The raw data has been supplied as Supplemental Dataset Files.

### Supplemental Information

Supplemental information for this article can be found online at http://dx.doi.org/10.7717/peerj.2686#supplemental-information.

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
