# Peer review of "Reanalysis of the anthrax epidemic in Rhodesia, 1978–1984"

_PeerJ, doi:10.7717/peerj.2686_

## Round 0.1 · original submission · Major Revisions

Dear authors,

Thank you very much for the opportunity to review your work. After reading the text and reports of the reviewers, I think it has scientific merit to be published once some issues are solved. Therefore my decision is MAJOR REVISION.

With respect and warm regards,
Dr Palazón-Bru (academic editor for PeerJ)

·

Basic reporting

Adequate and compliant with the criteria list.

Experimental design

Adequate and compliant with the criteria list.

Validity of the findings

Adequate and compliant with the criteria list.

Additional comments

Line 122: 2 out of 6 are too low to be presented in %.

Line 266-267: Sentence not clear

Line 275: I believe that "more" is to be preferred to "greater"

Figures 1, 2, 4, 5, 6:
a. A line graph is inadequate to present the information since each data point is represented by a discrete number and line graphs should be used to present continuous data. Consequently a bar graph would be more adequate.
b. I suggest adding color to the graphs (they may be already colored but I got a b/w version)..

Figure 3:
a. Are the data points averages of the years 1960-1980? I f so please note this in the caption.
b. For clarity's sake, please add 2 arrows to show the starting point of the animal and human outbreaks.

Reviewer 2 ·

Basic reporting

No Comments.

Experimental design

No Comments.

Validity of the findings

No Comments.

Additional comments

The authors performed a historical reanalysis of the data to describe the spatiotemporal patterns of the 1978–84 anthrax epidemic in Rhodesia. They have explored the spatial distribution dynamics of the anthrax outbreaks in humans from November 1978 to October 1980, and revealed the evaluated geographic involvement and numbers of human and cattle cases. In general, this study contains helpful information to understand the epidemiological features of this epidemic of anthrax in Rhodesia, and could give a clue to future studies on the etiology of this epidemic.

My major comments:
(1) It is not clear how data were collected and processed in the Materials and Methods section, and maybe the literatures needed to be cited here according to their categories such as historical (pre-epidemic) data, route of infection, severity of disease, meteorological data and seasonality, host animals, potential vectors, and vaccination coverage.
(2) Geospatial analysis needs to be described in more detail, e.g., How to identify an “outbreak” (how many cases)? How to conduct the analysis if two or more epidemic waves occurred in a hospital? The mean of outbreak coordinates seems not fitting for sporadic outbreaks in a wide area, maybe the map series of number of cases for each month or three months are optional.

Minor comments:
(1) Lines 124-125: No human or bovine anthrax cases were reported in all of Rhodesia from 1972 to 1977. However, it seems human cases were reported and displayed in the Figure 1 and limited outbreaks in bovines or humans were reported in “Mhondoro Tribal Trust Land (1974)” (Lines 120-121 ). Please check literatures and correct it.
(2) Lines 205-207: “Resource strain at the hospital was reflected in demand for hospitalizations, was noted in January 1980 and again in July 1980, which was the result of two waves of patients who were hospitalized for two to five weeks.” which is not consistent with Figure 6.
(3)The name of each province needs to be displayed in Figure 7, which is helpful for checking the locations of them.
(4) Table 1: “The first geospatial time cluster identified in the data was September 1979, which was statistically significant.” However, the p-value (0.055) is >0.05, it is also not statistically significant.
(5) The panels in Figure 7 seem unbalanced in time intervals, some are monthly, and some are three months.

Reviewer 3 ·

Basic reporting

The authors state that they searched for all literature on anthrax in southern Africa, however, there are glaring omissions in their search:
*Temporal and spatial distribution of cattle anthrax outbreaks in Zimbabwe between 1967 and 2006
*Spatial modelling of Bacillus anthracis ecological niche in Zimbabwe
When including these two omitted pieces of literature it appears the epidemic can attributed to a wide geographic area that is suitable for the presence of Bacillus anthracis. The authors current analysis does not support their conclusion of a geotemporal spread.

Experimental design

Methodological issues need to be addressed in the SaTScan analysis. Anthrax is a seasonal disease. Therefore, the use of the SaTScan on a monthly scale needs to be adjusted for seasonality. I suggest a review of the SaTScan manual. Regardless, the reporting of clustering results above p=0.05 are rather meaningless.

Validity of the findings

The authors state that they searched for all literature on anthrax in southern Africa, however, there are glaring omissions in their search:
*Temporal and spatial distribution of cattle anthrax outbreaks in Zimbabwe between 1967 and 2006
*Spatial modelling of Bacillus anthracis ecological niche in Zimbabwe
When including these two omitted pieces of literature it appears the epidemic can attributed to a wide geographic area that is suitable for the presence of Bacillus anthracis. The authors current analysis does not support their conclusion of a geotemporal spread.

The authors do not discuss how different climatic zones with latitudinal variation in precipitation may influence the timing of outbreaks. A hypothesis of the timing of anthrax outbreaks states the timing of outbreak is related to variation in precipitation patterns and temperature. While the authors provide national climatic data this is of little use given the small spatial scale used in the analysis.

Additional comments

This manuscript revisits an outbreak of anthrax, an important neglected zoonotic disease found nearly worldwide. The manuscript is interesting although there are several issues (methodological and others) that need to be addressed.

The use of hospital locations for enumerating the spread of anthrax is problematic for several reasons: access to facilities is going to vary causing individuals, particularly those in rural agrarian occupations, to travel far distances to seek treatment, and individuals may not choose the closest hospital if they believe a better facility exists. Rather a better use is to identify the community of origin as the putative source of infection (likely not available). Regardless the use of data at this resolution does not support a geo-temporal spread of the disease.
The etiological agent of anthrax was not confirmed until the 1880’s, so discussions of a lack of reporting dating back to the early 1800’s is erroneous. Anthrax to this date is severely underreported despite advances in diagnostic and surveillance. So it is likely that historical accounts of reporting in humans and particularly wildlife will be wildly inaccurate.

---

## Round 0.2 · accepted · Accept

Dear authors,

All the reviewers have indicated that your manuscript has high standards to be published in PeerJ. In other words, your research has great scientific merit.

Congratulations!

With respect and warm regards,
Dr Palazón-Bru (academic editor for PeerJ)

·

Basic reporting

No comments

Experimental design

No comments

Validity of the findings

No comments

Reviewer 2 ·

Basic reporting

No Comments.

Experimental design

No Comments.

Validity of the findings

No Comments.

Additional comments

The authors have considered my feedback and successfully addressed my concerns. I have no further comments.

Reviewer 3 ·

Basic reporting

The authors have, for the most part, adequately addressed my comments and suggestions.

Experimental design

No Comments

Validity of the findings

No Comments